# Sensory Characteristics and Volatile Organic Compound Profile of Wild Edible Mushrooms from Patagonia, Argentina

**DOI:** 10.3390/foods13213447

**Published:** 2024-10-29

**Authors:** Carolina Barroetaveña, Gabriela C. González, Eva Tejedor-Calvo, Carolina Toledo, Maria B. Pildain

**Affiliations:** 1Consejo Nacional de Investigaciones Científicas y Técnicas (CONICET), Godoy Cruz, Buenos Aires 2290, Argentina; cbarroetavena@ciefap.org.ar (C.B.); ctoledo@ciefap.org.ar (C.T.); mbpildain@ciefap.org.ar (M.B.P.); 2Área de Fitopatología y Microbiología Aplicada, Centro de Investigaciones y Extensión Forestal Andino Patagónico (CIEFAP), Ruta Nacional 259 Km 16, Esquel 9200, Argentina; 3Engineering Faculty, Universidad Nacional de la Patagonia S. J. Bosco, Ruta 259 Km 4, Esquel 9200, Argentina; 4Department of Plant Science, Agrifood Research and Technology Centre of Aragon (CITA), Av. Montañana, 930, 50059 Zaragoza, Spain; 5Laboratory for Flavor Analysis and Enology (LAAE), Department of Analytical Chemistry, Universidad de Zaragoza, C/Pedro Cerbuna 12, 50009 Zaragoza, Spain; 6Natural and Heatlh Science Faculty, Universidad Nacional de la Patagonia S. J. Bosco, Ruta 259 Km 4, Esquel 9200, Argentina

**Keywords:** sensory properties, quantitative descriptive analysis, trained panel, food preservation methods, gastronomy

## Abstract

The Andean–Patagonian forests of South America offer a great variety of wild edible mushrooms, many with ancestral use and others linked to new mycogastronomic offers. However, their sensory properties and detailed characterizations have not yet been deeply explored and described, nor have their alterations due to cold storage. The aims of this work were to perform a sensory characterization through a trained panel evaluation, perform target volatile compounds analysis and evaluate post-harvest preservation methods effects on nine species of wild edible mushrooms with different trophic habits (*Cortinarius magellanicus*, *Panus dusenii*, *Fistulina antarctica*, *F. endoxantha*, *Gloeosoma vitellinum*, *Grifola gargal*, *Lepista nuda*, *Ramaria patagonica*, and *Cyttaria hariotii*). The sensory description of dehydrated specimens through quantitative descriptive analysis showed that panelists were a significant source of variation; *F. antarctica* and *R. patagonica* registered distinct sweet flavor/spice odor and wood/sweet flavor, respectively, and different textures. Refrigeration produced a rapid loss of sensory characteristics, whereas freezer conservation satisfactorily maintained the characteristics in *F. anctartica*, *R. patagonica*, *G. vitellinum,* and *C. hariotti* for at least four months. A total of 60 target volatile organic compounds were detected, corresponding to grass, mushroom, alkane, and pungent odors in *F. anctartica*, *R. patagonica*, and *G. vitellinum*. The detailed sensory characterization and post-harvest conservation options of these novel products constitute crucial information to promote their sustainable use and local development through innovative activities linked to tourism, such as mushroom gastronomy and mycotourism.

## 1. Introduction

Wild edible mushrooms (WEM) are one of the most diverse and abundant non-wood forest products from the Andean–Patagonian region [1]. They have played a crucial role in the indigenous diet since prehistoric times, as well as in present populations [2,3,4]. In particular, the ancestral consumption records of *Fistulina antarctica* Speg. (n.v. cow tonge), *Ramaria patagonica* (Speg.) Corner (n.v. changle) and the newly consumed *Gloeosoma vitellinum* (Lév.) Pat. (n.v. stick ear). The three of them, widespread along Nothofagaceae forests, are among the most used endemic WEM by regional chefs [5].

WEM consumption has increased worldwide in recent years since they have a promising potential in future forest-based value chains, contributing to human nutrition and gastronomy [6,7]. These resources have gained special interest in recent decades because of their bioactive compound contents and potential to diversify and boost local economies [4,8,9,10]. Their culinary and commercial value is mainly derived from their sensory and nutritional properties. In addition to having particular and varied odors, flavors, colors, shapes, and textures, they are considered healthy food because of their low fat and sodium contents and high protein content, with a significant amount of essential amino acids with better nutritional quality compared with vegetables [11,12,13]. These qualities make them very suitable for specific diets such as: low cholesterol, low sodium, and vegetarian/vegan or in situations of scarce availability of animal proteins. Moreover, the presence of various bioactive substances offers multifunctional medicinal properties, such as antioxidants or antimicrobials [14,15,16]. In Patagonia, different regional protocols for good harvest practices and food safety; the inclusion of 21 species in the Argentine food code (chapter 16, art 1249); establishing agreements for the implementation of mycotourism paths; training in the identification and safe consumption, among others, are activities that strengthen their sustainable use and encourage their consumption [17].

The availability of wild mushrooms in Patagonia is conditioned by their marked seasonality, along with a highly variable frequency of discovery [18]. Their very high water content (most around 90% of fresh weight) and the exposure to spoilage damage (physical, microbial, and mechanical) during the post-harvest period determine their very short shelf-life [19,20,21,22]. Thus, the food industry looks for suitable and feasible conservation techniques that maintain the sensory characteristics of mushrooms during their distribution and commercialization. Drying and freezing are the most common methods for preserving mushrooms [23,24]. Drying is the cheapest preservation method [25] and the most widely used in Patagonia [5], as it is easily available in rural harvest areas. It significantly increases the shelf-life of mushrooms, which can be safely stored in proper conditions, preventing most of the spoilage, such as enzymatic browning, microbial growth, and so on, by reducing water activity [26,27]. As reported for *Lentinula edodes* (Berk.) Pegler (Shiitake) by Subramaniam et al. [27], there also exists a consumer/chef preference for dried mushrooms in Patagonia because they have richer odorthan fresh ones. This could be partly due to the umami taste produced by Maillard reaction products, which occurs during thermal dehydration [28], but also to the concentration of compounds due to water loss. On the other hand, fresh-freezing and freeze-drying are other postharvest preservation methods. González et al. [29] demonstrated that these last two methods are more efficient in preserving total phenol contents and bioactivity than thermal dehydration for *R. patagonica* and *F. antarctica*.

Trained panels using Quantitative Descriptive Analysis (QDA^®^) are an appropriate tool for determining the sensory properties of food products. In addition, sensory evaluation of post-harvest preservation methods is essential in food quality control and technology, allowing the establishment of patterns that can be used at any time to describe and analyze a product [30]. However, no studies have been conducted reporting sensory properties for preserved WEM from Patagonia native forests.

Despite the high worldwide consumption of mushrooms, few studies have concerned their odor. Fungi emit a wealth of highly diverse volatile organic compounds (VOCs) [31], which play central roles in fungal interactions. The typical “mushroom-like flavor” is attributable to several water-soluble substances, such as 1-octanol, 1-octen-3-one or 3-octanone. The determination of the volatile profile has been used to compare strains or species [32], to relate ecological functions and lifestyles of individual fungal species [33,34], and to assess the authenticity of flavoring substances and food products commercially available [35]. In addition, distinctive odors have been used as taxonomic markers for mushroom species identification [33]. Nevertheless, fungi have been relatively understudied compared with that of other living organisms [36], and the need to generate VOC information for wild edible fungi in forest environments is even more noticeable.

This study used dehydrated batches and a trained panel to generate a sensory attribute description for the nine most abundant and/or paradigmatic wild edible mushroom (WEM) species from Patagonia. Additionally, it sought to evaluate the influence of two post-harvest methods (refrigeration and blanching/freezing) on the sensory shelf life of these WEM species. Finally, this study analyzed the volatile organic compounds (VOCs) profile of selected edible species with different trophic habits (ectomycorrhizal and wood-degrading) and sensory properties to identify potential key aromatic compounds.

## 2. Materials and Methods

### 2.1. Wild Edible Mushroom Samples

Different WEM from Patagonia were harvested between 2015–2016. Among them are *Gloeosoma vitellinum* (formerly *Aleurodiscus vitellinus*), *Cortinarius magellanicus* Speg., and *Panus dusenii* Bres. (Formerly *Hydropus dusenii*), *Fistulina antarctica*, *F. endoxantha* Speg., *Grifola gargal* Singer, *Lepista nuda* (Bull.) Cooke, *Ramaria patagonica*, and *Cyttaria hariotii* E. Fisch were obtained along the fruiting season from native Nothofagaceae forests from Chubut, Río Negro, and Neuquén provinces (Argentina) (Table 1). This collection’s taxonomy was certified in fresh, following Singer [37], Horak [38], Valenzuela Flores [39], and Rajchenberg [40] publications. In addition, one representative collection of each species was herborized and deposited in the Herbarium of the Andean Patagonian Forest Research and Extension Center (HCFC).

Nine collections of fresh specimens (10–15 samples per species) of the studied species were used for each evaluation (except *G. gargal*, where only 5 samples were analyzed given the low number of available specimens, determined by its low frequency of fruiting and restricted geographic habitat). Each collection was conditioned, removing remains of humus and soil (without separating the pileus from the stipe when present).

### 2.2. Quantitative Descriptive Analysis for Sensory Attribute Description

As dehydrated mushrooms are the most frequent way they are stored and commercialized in Patagonia, the dehydration process was carried out to preserve mushroom species and avoid degradation during the sensory analysis. Samples of fresh mushrooms of each species were cut into 0.5 cm slices (depending on species sizes; species of small size and thin flesh were dried completely; regarding stipitate species, only *L. nuda* stems were removed) and placed in a dehydrator at 40 °C (Numak, DF-71L, Zelian, Buenos Aires, Argentina) until constant weight. They were rehydrated in mineralized water at room temperature (20–24 °C) for 2 h. The soaking water was then carefully discarded, and another quick wash was performed to remove any remaining grit before sensory evaluation. The panel was constituted by 10 untrained people (5 male and 5 female) after recruiting individuals capable of using both terms and expressions needed for the activity. All participants provided informed consent to partake in this study. The appropriate protocols for protecting the rights and privacy of all participants were used. CIEFAP Research Center gave the permission to conduct sensory panel research. Testers were trained following ISO 13301:2022 (sensory analysis: general guidance for measuring odor, flavor, and taste detection thresholds by a three alternative forced-choice procedure) [41] and 5495:2006 (sensory analysis: initiation and training of assessors in the detection and recognition of odors) [42]. This study was conducted according to the ethics guidelines of the Nagoya Protocol (Argentinian National Law N° 27246). Panelists carried out a total of 3 sessions to describe 3 species each time [43]. The training sessions consisted of 3 different sessions of two hours each (spaced every 15 days), evaluating 3 species per session. Each session was organized in three stages: (1) individual description of the attributes (odor, flavor, and texture) for each species using different descriptors (species samples were randomly assigned to each panelist); (2) Agreement on the identified descriptors, by discussing to reach agreement on the identified descriptors and building a single consensus list, including descriptors with a citation frequency equal to or greater than 6 [44], for each of the evaluated attributes of each sample; (3) Once the basic lexicon/final list of attributes was established, final testing was conducted on each mushroom sample. Samples were tested individually in random order within each of two replicates using the 0–10 scale [41], where 0—no perception of the descriptor; 1 to 4—minimally perceived descriptor (soft); 5 to 7—moderately perceived descriptor (moderate); 8 to 10—maximum intensity of perception of the descriptor (strong). The samples were only tasted and then spit out by the panelist after the description. Mineralized water, unsalted crackers, and carrots were provided to cleanse the palate during testing.

Each sample was evaluated in duplicate, following a completely randomized experimental design. Two-way ranking analysis of variance (ANOVA) was performed for each group of samples and panelists. The means were separated according to Tukey’s a posteriori test (*p*-value < 0.05) to detect significant differences among samples and panelists for the sensory variables studied. Statistical analyses of sensory evaluation were performed with the statistical package InfoStat version 2011 [45]. The Principal Component Analysis (PCA) with sensory evaluation data was performed with the PanelCheck program V1.4.2.

### 2.3. Evaluation of Refrigeration and Blanched/Frozen on Sensory Shelf Life

First, an initial fresh sensorial description was carried out. Fresh collections of each species were sensorial and characterized by the authors, including odor, texture, flavor, and color, to use them as a reference for later comparisons with the conservation treatments.

Method I (refrigeration): mushrooms (from each initial 10-sample batch) were cut into slices, 0.5 cm thick depending on specimen sizes, placed in three 15 × 27 cm^2^ polystyrene trays, covered with polyethylene film (Film HD STRETCH) and placed in a refrigerator at 4 °C. Trays stored in the refrigerator were evaluated once a week until signs of degradation were detected through sensory attribute loss.

Method II (blanched and frozen): mushrooms (from each initial 10-sample batch) were cut into 3–4 cm thick pieces (depending on species sizes), submerged in a boiling water bath for 2 or 3 min (blanching), immediately cooled under a stream of cold water, drained and placed in polystyrene trays, covered with polyethylene film and placed in the freezer at −18 °C. Trays were examined at 4, 6, and 8 months after being thawed, verifying the loss or variation in the reference sensory attributes, and then discarded.

To quantify the magnitude of attribute loss (odor, flavor, texture) for method I and II, a numerical scale (0–3) was used: 0—null: the material retains all the sensory characteristics (odor, flavor, texture and color) recorded on the fresh material; 1—mild: a slight loss of the sensory characteristics (odor, flavor, and texture) is detected, and the color is preserved; 2—moderate: a moderate loss of sensory characteristics (taste and texture) is detected, including color and a slight putrid odor; 3—severe: a severe loss of sensory characteristics (taste and texture), including color, and a strong putrid odor is detected. Putrid odor refers to any unpleasant smell caused by amines, sulfur compounds, aldehydes, or carboxylic acids associated with the senescence process.

### 2.4. Volatile Organic Compound Analysis by SPME-GC-MS

The volatile organic compounds (VOCs) of three selected mushroom species with different trophic habits and sensory qualities (detected by the panel and the post-harvest evaluation) were analyzed: *Ramaria patagonica* (mycorrhizal), *Gloeosoma vitellinum* (thin branches or trunks on live or dead wood, wood rotter in a wide range of *Nothofagus* species), and *Fistulina antarctica* (stem wood decayer), in order to deeply investigate their aromatic compounds and related them with the odor noticed by the trained panel.

The methodological approach was based on works carried out by Tejedor-Calvo et al. [35]. A solid-phase microextraction (SPME) was used to extract the aromatic compounds. For that, a fused silica fiber coated with a 50/30 mm layer of divinylbenzene/carboxen/polydimethylsiloxane from Supelco (Barcelona, Spain) was chosen. The samples were freeze-dried (BK-FD10 Series, BioBASE, Bariloche, Argentina) at a condenser temperature of −64 °C, and the vacuum was kept at 20 Pa. Then, the samples were ground, and sieved to obtain a particle size lower than 0.5 mm and stored in darkness at −20 °C until further use. Later, 2 g of sample were placed in a 15 mL glass vial closed with a septum. After the vial was conditioned at 50 °C for 10 min. The fiber was then exposed to the headspace of the vial for 20 min. The VOC analysis was carried out by duplicates in each sample.

The VOC profile of the different samples was analyzed using a gas chromatograph Perkin Elmer Clarus 600 Series coupled with a Perkin Elmer Clarus 600 mass spectrometer detector (Chatsworth, CA, USA). This SPME-GC-MS instrument was equipped with a capillary column HP-5MS of 30 m, 0.32 mm i.d., 0.25 μm film thickness, and a flow of 1 mL/min with helium as a carrier gas. The samples were injected in splitless mode. The oven temperature was 45 °C held for 2 min, 45–200 °C at a rate of 4 °C/min, and finally to 225 °C at 10 °C/min, and held for 5 min. The MS used the electron impact mode with an ionization potential of 70 eV and an ion source temperature of 200 °C. The interface temperature was 220 °C. The MS scanning was recorded in full scan mode (35–250 *m*/*z*). A TurboMass software version 6.1 was used for controlling the GC-MS system.

Data treatment was carried out according to Tejedor-Calvo et al. [35]. Peak identification of the VOCs was achieved by comparison of the mass spectral with mass spectral data from the NIST MS Search Program 2.0 library and by comparison of previously reported Retention Indexes (RI) with those calculated using an n-alkane series (C6–C20) under the same analysis conditions. The n-alkane series and standards for MS identification (all standards of purity higher than 95%). A semi-quantification was achieved by integrating the area of the main ion of each compound and normalization by calculating the relative percentage using the OpenChrom^®^ (V. 1.5.0) program. This allowed the comparison of each eluted compound between samples.

## 3. Results

### 3.1. Sensory Descriptors and Selected Attributes from Rehydrated Samples

A total of 65 descriptors were mentioned in three sessions, of which 38.4% corresponded to aroma, 36.9% to flavor, and 24.6% to texture attributes (Table 2). Each group of samples was formally evaluated by scoring only the most frequently cited descriptors in those sessions (shown in Table 2); it was estimated that the higher the frequency, the greater the consensus on the usefulness of a certain attribute to describe the sensory characteristics of the samples.

Results from the formal panel sessions, analyzed using analysis of variance, showed that the panelists were a significant source of variation for most of the descriptors of the three evaluated attributes (*p*-value < 0.05). Likewise, it was possible to verify that there were significant differences between the descriptors for the evaluated species (*p*-value < 0.05).

In the sensory analysis of the odor, average intensity values between mild and moderate were obtained for the mushroom descriptor, except for *L. nuda,* for which its perception was below 1 (Appendix A); *Fistulina antarctica* and *C. magellanicus* presented the significantly highest intensity values, followed by *F. endoxantha*, *P. dusenii,* and *R. patagonica*, with moderate intensity values (between 5 and 6; Appendix A). For *G. gargal*, the almonds odorand flavor descriptor [45] were one of the most frequently cited by the panelists, although with soft intensity but statistically significantly higher compared with the other species (Appendix A).

The “mushroom” flavor was another descriptor that allowed for establishing differences between species, with mild intensity values for *L. nuda*, *G. vitellinum,* and *P. dusenii*. Woody flavor also showed significant differences between *R. patagonica* and *L. nuda*, with moderate intensity in the first case and below 1 in the second. *Ramaria patagonica*, *F. antarctica,* and *F. endoxantha* obtained the highest average intensity values for sweet flavor compared with the rest of the species (Appendix A). The bitter taste was not included in the quantitative analysis of the low frequency reached by the panelists. Results for texture were generally moderate to mild in average intensity. A fleshy texture was the only factor that obtained significant differences between species, reaching a moderate intensity in *L. nuda* and a mild intensity in *G. vitellinum*, *F. antarctica*, and *F. endoxantha*. In addition, significant differences were found for the cartilaginous descriptor between *L. nuda* (low value: 0.4), *R. patagonica,* and *P. dusenii* (high values: 3.8 and 3.05, respectively). *Lepista nuda* and *R. patagonica* showed significant differences for the “hard” descriptor, with moderate average intensity for the former and close to zero for the latter. Fresh specimens of *L. nuda* were characterized by a fleshy texture that is lost after dehydration. Significant differences between samples were recorded for soft texture, with moderate average intensity for *G. vitellinum* and *C. hariotii*. *Cortinarius magellanicus* was characterized by a soft mucilaginous texture, which differed significantly from *G. vitellinum* and *C. hariotii* with an almost zero average intensity.

The average descriptor values for each species (Figure 1) constitute a sensory profile not reported previously that allows for the differentiation of wild edible mushrooms, discriminating between those with the best aptitudes for consumption. *Gloeosoma vitellinum*, *C. hariotii*, *R. patagonica*, *P. dusenii*, *C. magellanicus*, *F. antarctica*, and *F. endoxantha* presented a mild to moderate fungal aroma. A mild fungal flavor was reported in *G. vitellinum*, *L. nuda*, *C. hariotii*, and *P. dusenii*. *Ramaria patagonica*, *F. antarctica*, and *F. endoxantha* were among the species with mild to moderate sweet flavor. *Grifola gargal* and *C. magellanicus* showed a mild almond and nut flavor, respectively. The texture was variable between the studied species. *Fistulina antarctica*, *F. endoxantha,* and *L. nuda* were characterized by a soft to moderate fleshy texture, *C. magellanicus* with soft mucilaginous, and *C. hariotii* and *G. vitellinum* with soft. *Ramaria patagonica* and *P. dusenii* were cartilaginous, while *G. gargal* presented a leathery texture.

The PCA analysis used to explore the possible correlations between sensory properties and mushroom species (Figure 2) explained 49% of the data variability with the two first components, indicating a complex correlation between the sensory attributes detected by the panelist and the mushroom species. The sample distribution clearly separated *G. gargal* from the other species (green ellipse). Apart from the almond odor and flavor reported by Rajchenberg [46] the *G. gargal* sample was characterized by a leathery attribute. Among the rest samples, two different groups were observed. Samples placed below the X axis (blue ellipse) (*C. magallanicus*, *F. endoxantha*, *P. dusenni*, *F. antartica,* and *R. patagonica*) were characterized by mushroom odor and sweet flavor, whereas those located on the top right of the plot (yellow ellipse) (*C. hariotti*, *L. nuda*, and *G. vitellinum*) by a nutty odor but also mushroom flavor. In between these two groups, some sensory attributes were placed (earthy odor, sweet odor, spices flavor, and wood odor) indicating that those attributes were not representative of any of the species. Apart from mushroom odor and flavor, the other attributes (odor and flavor) were located close, indicating a close relationship between odor and flavor for the same attributes, i.e., spices, sweet, wood, nuts, and almonds.

### 3.2. Sensory Characteristics Under Cold Preservation Methods

Under refrigeration (Method I), *P*. *dusenii* and *R. patagonica* preserved their sensory properties between 7 and 8 days, *G. vitellinum*, *L. nuda*, and *C. hariotti* between 4 and 6 days, while *C. magellanicus*, *F. antarctica*, and *F. endoxantha* lost their attributes after 4 days (Table 3). Color and odor were the first parameters to differentiate it from fresh material (Appendix A).

The conservation of up to 8 months of blanched and frozen material (Method II) showed slight changes for *G. vitellinum*, *P. dusenii*, *C. hariotii*, and *L. nuda*, and moderate for *F. antarctica* and *C. magellanicus*. Instead, for *F. endoxantha* and *R. patagonica,* those changes were registered at 6 and 4 months, respectively (Table 3). Regarding color, excepting *C. magellanicus* and *F. antarctica* which showed a notable loss of their violet and reddish hue, respectively. The rest of the species retained their original color after being subjected to both methods (refrigeration and blanched/frozen). Dissimilar to the material stored in the refrigerator, the specimens stored at −18 °C did not show signs of degradation; however, a slight loss of attributes was observed. The texture was the most affected in seven of the evaluated species, except for *C. magellanicus* and *C. hariotii* (Table 3); a loss of odor (in *G. vitellinum*, *P. dusenii*, *L. nuda*, and *C. magellanicus*) and flavor (*G. vitellinum*, *P. dusenii*, *R. patagonica*) were also noticeable (Appendix A).

### 3.3. VOCs Study

The VOCs of three mushroom species were deeply studied (*R. patagonica*, *Gloeosoma vitellinum,* and *F. antartica*) using SPME-GC-MS. Mushrooms were selected according to the differences observed in the sample distribution of the sensory analysis PCA plot (Figure 2) and their habitat type (Figure 1).

Among the VOCs with odor properties (Table 4), hexanal (grass odor) (17%) was clearly the main compound in *R. patagonica*, followed by 2, 6-dimethyl-nonane (no odor). *G. vitellinum* showed a higher number of aromatic compounds: 3-octanone (mushroom odor), 2-undecanone (orange, fresh odor), and some alkanes such as undecane (16.9%), dodecane (6.2%), and pentadecane (13.9%). As in the other mushrooms, pentadecane (alkane odor) (15.6%) was one of the main volatile compounds of *F. antarctica*, together with 2-ethyl-1-hexanol (no odor) (14.7%). Some minor compounds were found in similar percentages in all the samples: 2-methyl-nonane (1.9–3.0%), 2,5-dimethyl-nonane (2.2–3.7%), 2-methyl-undecane (1.7–2.7%), Ethyl-1-hexanol acetate (1.7–2.7%), 2,6-dimethyl-undecane (3.2–3.9%), tetradecane (1.4–2.5%), 2-undecanone (2.2–3.2%), and 4,6-dimethyl-dodecane (3.0–4.6%). Most of them are not characterized as being odorous; however, 2-undecanone is described with an orange, fresh, green odor.

Those odorous compounds listed in Table 4 were studied to compare the different aromatic profiles in WEM (Figure 3). Although the alkane odor reached up to 22.0, 38.3, and 29.7%, respectively, in the WEM species studied (*R. patagonica*, *G. vitellinum*, *F. antarctica*), the odor descriptor is not very clear as for example, mushroom odor. Therefore, to better observe those odor descriptors well-defined, the odor descriptor alkane was removed from the graphics. The WEM study reported different aromatic profiles, with *R. patagonica* being the simplest, with only four odor descriptors detected. This species was characterized by a grass odor(17.2%), followed by citrus (3.4%), fat (2.4%) and mushroom (1.1%). Similar values for citrus and fat odor were found in *G. vitellinum*; however, its profile was more complex (six odor descriptors) than the other species. The main aromas were pungent and mushroom (4.0 and 2.6%, respectively). *F. antarctica* showed a very distant profile from the other two species; in fact, fruity (7.5%) and malt (2.5%) odor compounds were only detected in this species, although it shares mushroom (2.7%) and sweet content (1.5%) with *G. vitellinum*. Therefore, we can differentiate the WEM studied for more grass and fruity odor for *R. patagonica* and *F. antarctica,* respectively, and a more complex profile (mix between pungent and mushroom) for *G. vitellinum*.

## 4. Discussion

Wild edible mushrooms are increasingly recognized due to their content of bioactive compounds and their potential to diversify local economies. The previous literature indicates that their culinary and commercial value is mainly due to their sensory and nutritional properties [47]. Thus, the food industry looks for suitable and feasible conservation techniques that maintain the sensory characteristics of mushrooms during their distribution and commercialization. WEMs are used in specialized custom-made products and are increasingly marketed as well-being products embedded in recreation or educational services or as products that include experiential services such as guided tours, fairs, or events [48]. However, sensory evaluation of post-harvest preservation methods is not available for preserved WEMs from Patagonia native forests. Here, we aim to systematically close this gap and establish sensory attribute descriptions linked to the sensory shelf life of different post-harvest methods and odor with the analyses of the VOC profile.

Different thermal preservation methods were applied to this novel WEM species to explore the most usual and accessible techniques applied to mushrooms in the food industry in order to widen options for their commercialization. Even though in terms of quality, energy consumption, and shelf life, vacuum cooling is more efficient, the high investment costs and the greater weight loss compared with that seen under conventional cooling usually limit its application [47].

Textural quality is the sensory characteristic most negatively affected by cold preservation. This can be attributed to various factors such as denaturation, dehydration damage, drip loss, tissue fractures, and mechanical damage by ice crystal growth during freezing [46]. In traditional air-blast freezing systems, severe damage is caused to the texture due to large ice crystals (mostly extracellular) formed during slower block freezing [49,50]. Despite this inconvenience, freezing is one of the most popular and efficient methods for food preservation as it allows the best retention of nutritional and nutraceutical values [51]. This has been recently demonstrated by González et al. [29] for Patagonian mushrooms, where the fresh-freezing method exhibited the highest inhibition of free radicals and total phenolic compounds compared with other preservation methods. Phenolic compound liberation might be enhanced because of compound breaking and liberation during the freezing process. The differential antioxidant bioactivity potential of WEMs after freezing methods and its applications deserve further study.

Mushroom senescence, another sensory characteristic analyzed and affected by the preservation method, is related to increased enzymatic activity, such as phenol oxidases (tyrosinase, laccase) and cell wall enzymes (glucanases and chitinases), from the moment the product is harvested [52]. Indeed, firmness loss during the storage period may be the result of diminished chitin synthesis, chitin depolymerization, or both effects in combination [53]. Tejedor-Calvo et al. [54] remarked that chitins and β-glucans are the major constituents of fungal membranes, and they are responsible for the structural conformation and firmness in truffles (*T. melanosporum* and *T. aestivum*). This kind of mushroom contains up to 12% of chitin, while other species only reach 5–10% [11,55,56]. The preservation methods described were able to extend the optimal sensory attributes up to 4 days and 4 months with refrigeration and 4 months with blanched and frozen (excepting *R. patagonica* that lost texture and original wood odor in 6 months and *F. endoxantha* with severe texture lost at 8 months with freezing).

One of the central outcomes of this study is the qualitative analysis of dehydrated-rehydrated specimens. The panelists were a source of significant variation for most of the descriptors of the three evaluated attributes, possibly because different sectors of the scale were used. These results suggest that greater training in the use of correct scales is needed [57]. Notwithstanding this, we observed attribute differences for several WEM species. Regarding the odor descriptor, *Fistulina antarctica* and *C. magellanicus* presented the significantly highest intensity values, followed by *F. endoxantha*, *P. dusenii*, and *R. patagonica*, with moderate intensity. Considering the sensory profile (Figure 1), *F. antarctica* showed the highest mushroom odor in comparison with the rest of the species and a similar VOC profile to *G. vitellinum*.

In relation to flavor attributes, high sweet intensity values were detected in *Ramaria patagonica*, *F. antarctica*, and *F. endoxantha*. This might be due mainly to the sugar content (*R. patagonica:* 0.86 mg/100 g fructose, and 8.60 mg/100 g mannitol; *F. antarctica:* 10 mg/100 g fructose, 1 mg/100 g mannitol, 8 mg/100 g trehalose; *F. endoxantha:* 23 mg/100 g fructose, 3.70 mg/100 g trehalose [16]), but some volatiles classified as sweet or fruity could be involved in the *F. antarctica* sweet flavor. Some alcohols and esters can have sweet aromatic profiles. For example, 1-octen-3-ol has been described as mushroom alcohol with a strong, sweet, and earthy odor [58] present in *Tricholoma matsutake* [59], *Flamulina velutipes* [60], *Lentinula* spp. [61], and *Boletus* spp. [62]. The *R. patagonica* sensory profile showed a high percentage of grass odor in comparison with the other two mushrooms, and it can be related to those volatiles with malt and green aromatic properties. The mushroom flavor was higher in *G. vitellinum,* so the tasting phase might enhance the volatile perception, probably because of the mouth temperature that increases compound volatility. Chen et al. [28] reported that mushroom flavors are influenced by the presence of three types of compounds, including sweet type (high content of alanine, glycine, and threonine), monosodium glutamate type (with high levels of aspartic acid and glutamic acid), and bitter type, probably due to some phenolic compounds.

Regarding the VOC profiles of the analyzed WEM, we can differentiate a more grass and fruity odor for *R. patagonica* and *F. antarctica,* respectively, and a more complex profile (mix between pungent and mushroom) for *G. vitellinum*. Registered VOCs as hexanal (grass odor) or 3-octanone (mushroom odor) also were previously described in four cultivated mushrooms (*Agaricus bisporus* sp. *bisporus, Agaricus bisporus* sp. *brunnescens, Lentinula edodes, Grifola frondosa*): 1-hexanol (resin, flower, green odor), hexanal (grass, tallow, fat odor), 3-octanone (mushroom odor), 1-octen-3-ol (mushroom odor), and 3-methylbutanal (malt odor) [60]. In addition, truffles contain these molecules, specially 3-methylbutanal in higher amounts (62.3 mg/100 g truffle) and 1-octen-3-ol in lesses amounts (2.2 mg/100 g truffle) [35], and their content amount is also related with heating preservation treatments (sterilization or pasteurization) [6].

Comparing these results with those obtained by the trained panel, a tentative correlation might be established. However, it is necessary to consider that the attributes selected by the trained panel might not be the same as those attributable to the chemical compounds. For instance, grass odor in the *R. patagonica* profile might be due to wood or sweet flavor (highest percentage in Figure 1) by the trained panel. The mushroom flavor detected by the trained panel was also in the VOC profile (Figure 3) as one of the highest odors detected. The *F. antarctica* profile described by the trained panel chose sweet odor and flavor as the main odorants, while more than half of the VOCs detected were described as sweet and fruity. In order to enhance this approach correlating VOCs and attributes, expert tasters training with standards and chemical compounds quantification would be required. Zhuang et al. [63] compared odor perception in *Boletus* species using instrumental and sensory techniques and concluded that the combination of these methods could suggest which volatile compounds may be responsible for the aromatic notes detected by the panel. However, establishing precise quantitative relationships between perceived odor and instruments remains a challenge. In general, the differences between the VOC analysis and the trained panel might be due to the perception threshold. Some molecules with low perception threshold despite their levels, might be difficult to detect by the human nose. However, some parameters such as solubility or temperature (as happens in the tasting phase of the sensory analysis) might increase the detection by tongue receptors. Therefore, a specific protocol to detect mushroom flavors and odors might be developed as other food products have (oil in blue glass or wine in tulip glass).

The widely studied Indigenous *Morchella* spp. and exotic *Suillus luteus* [64], followed by the native *Fistulina antarctica*, *Ramaria patagonica*, and *Cyttaria hariotti*, have been reported as the species with the greatest cultural importance at the regional level in Patagonia (Argentina); they were frequently mentioned for their commercial value, continuity of use over time, and outstanding sensory properties by Mapuche-tehuelche communities [3] and creole rural settlers, who are the main harvesters of WEM in the region.

## 5. Conclusions

With the use of QDA^®^ for sensory attribute description, evaluation for postharvest storage methods on sensory shelf life, and VOC analysis, we were able to identify shared and unique attributes, characterize the sensory losses given the preservation methods, and provide sensorial and odorous VOC profiles with standard and comparative methods for little-known Patagonian WEM. Currently, the studied species have occasional use. For that, the results of this work will help highlight their attributes and, therefore, contribute to the promotion of the use of these resources. The foster of mycotourism and mushroom gastronomy as sustainable identity and inclusive economic and educational activities that support local development, taking advantage of the outstanding regional tourist profile, are also key factors in the increased amounts of diversified, valued WEM. Thermal preservation methods with modern technology (freeze-drying and individual quick freezing) and chemical and physical postharvest treatments should be explored alongside the development of functional food products. Furthermore, it is important to incorporate forest management based on mycosilviculture and the generation of protocols for cultivating endemic saprophytic and wood-decaying species to increase harvest availability and market organization. Management and cultivation of WEM are strategically related to the increased, although sustainable, use of these wild resources.

## Figures and Tables

**Figure 1 foods-13-03447-f001:**
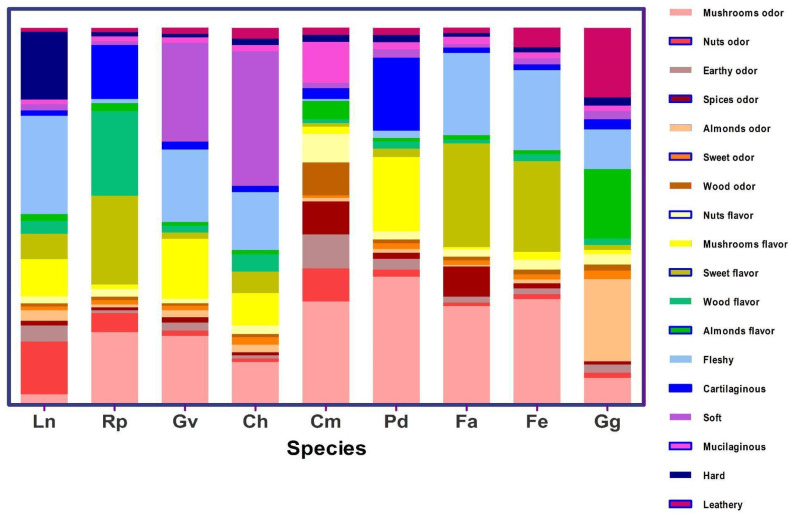
Sensorial profile (odor, flavor, texture) from the descriptive analysis of the dehydrated wild mushroom species: Ln: *Lepista nuda*; Rp: *Ramaria patagonica*; Gv: *Gloeosoma vitellinum*; Ch: *Cyttaria hariotii*; Cm: *Cortinarius magellanicus*; Pd: *Panus dusenii*; Fa: *Fistulina antarctica*; Fe: *Fistulina endoxantha*; Gg: *Grifola gargal*.

**Figure 2 foods-13-03447-f002:**
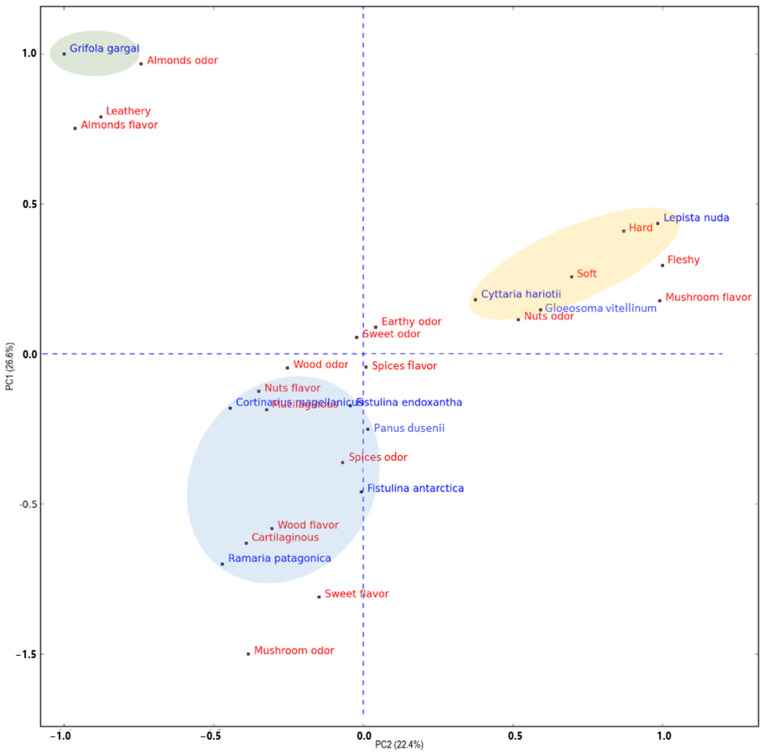
PCA biplot of 9 mushroom species (blue names) sensory attributes in the sensory analysis (red names). Color ellipses grouped mushroom species with similar sensory attributes.

**Figure 3 foods-13-03447-f003:**
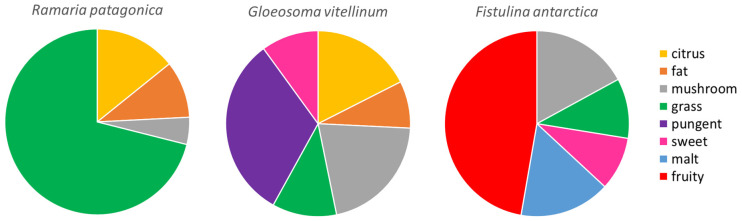
Odor attributes of VOCs detected in the most common wild edible mushrooms from Patagonia. Odor attributes correspond to those listed in Table 4. Data are expressed in percentages.

**Table 1 foods-13-03447-t001:** Physical description of wild edible mushroom species from Argentina. Pictures were taken by the authors. The consumption recommendations are from the Arts. 1249 and 1250 of the Argentinean Food Code.

Mushrooms Species	Picture	Shape	Basidiocarp Size	Habitat	Phenology	Consumption
*Gloeosoma vitellinum*	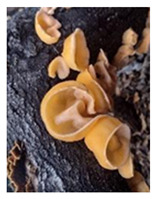	Discoid	2–6 cm × 4 cm	Wood	March to May	Complete fruiting can be consumed. Fresh or cooked.
*Cortinarius magellanicus*	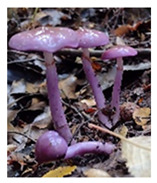	Pileus: flat or umbonate Stipe: equal	8 cm × 12 cm	Ectomycorrhizal	March to May	Complete fruiting can be consumed. Its rehydrated consumption is recommended.
*Panus dusenii*	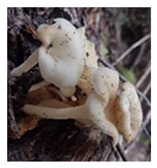	Pileus: umbilicate Stipe: equal	6 cm × 10 cm	Wood	March to April	Complete fruiting can be consumed. Cooked.
*Fistulina antarctica*	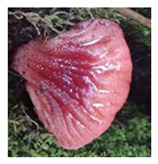	Dimidiate-flabellate or ungulate	15–30 × 10–20 × 3–10 cm	Wood	March to May	Complete fruiting can be consumed. Fresh or cooked.
*Fistulina endoxantha*	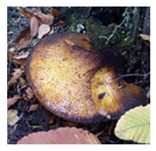	Diimidiate-flabellate or ungulate	15–30 × 10–20 × 3–10 cm	Wood	April to May	Complete fruiting can be consumed. Fresh or cooked.
*Grifola gargal*	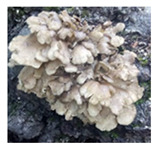	multipileate, rosette-like or cabbage-like, imbricated	up to 30 cm diam.	Wood	April to May	Complete fruiting can be consumed. Fresh or cooked.
*Lepista nuda*	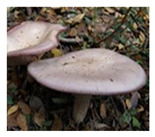	Pilus: flat or depressed Stipe: equal or club-shaped	5–15 cm × 10 cm	Saprophytic	March to May	Only the pileus can be consumed. Cooked.
*Ramaria patagonica*	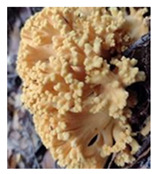	Coralloid	Up to 15 cm high	Ectomycorrhizal	April to May	Complete fruiting can be consumed. Cooked.
*Cyttaria hariotii*	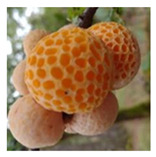	Globose	3–7 cm diam.	Parasitic	October to December	It is recommended to consume it fresh.

**Table 2 foods-13-03447-t002:** List of the 65 descriptors mentioned by the panel in the descriptive test on dehydrated samples. The most frequent citation is indicated. Samples were analyzed in triplicate.

Odor	Citation Frequency (%)	Flavor	Citation Frequency (%)	Texture	Citation Frequency (%)
Almonds	8.3 ± (0.01) *	Acrid	3.7 ± (0.22)	Soft	16.7 ± (0.02) *
Aniseed	1.0 ± (0.01)	Acidic	3.7 ± (0.32)	Fleshy	10.0 ± (0.45) *
Chocolate	4.2 ± (0.23)	Almonds	8.6 ± (0.13) *	Cartilaginous	13.3 ± (0.03) *
Crayon	1.0 ± (0.02)	Bitter	3.7 ± (0.24)	Corky	3.3 ± (0.01)
Sweet	9.4 ± (0.26) *	Astringent	2.5 ± (0.32)	Leathery	6.7 ± (0.35) *
Spices	7.3 ± (0.03) *	Chocolate	1.2 ± (0.01)	Crispy	3.3 ± (0.01)
Farinaceous	2.1 ± (0.12)	Sweet	8.6 ± (0.26) *	Hard	10.0 ± (0.01) *
Fermented	5.2 ± (0.34)	Spices	8.6 ± (0.28) *	Elastic	3.3 ± (0.15)
Floral	2.1 ± (0.12)	Farinaceous	4.9 ± (0.36)	Spongy	3.3 ± (0.13)
Citrus	1.0 ± (0.03)	Fermented	1.2 ± (0.02)	Fibrous	3.3 ± (0.023)
Damascus	5.2 ± (0.01)	Fruity	3.7 ± (0.04)	Firm	3.3 ± (0.034)
Nuts	9.4 ± (0.02) *	Dry nuts	7.4 ± (0.03) *	Mucilaginous	10.0 ± (0.21) *
Mushrooms	10.4 ± (0.15) *	Mushrooms	9.9 ± (0.02) *	Rubbery	3.3 ± (0.13)
Dry leaves	1.0 ± (0.02)	Greasy	3.7 ± (0.46)	Sticky	3.3 ± (0.21)
Wood	10.4 ± (0.13) *	Wood	8.6 ± (0.06) *	Slimy	3.3 ± (0.32)
Cooked mate	1.0 ± (0.01)	Honey	1.2 ± (0.21)	Doughy	3.3 ± (0.03)
Molasses	2.1 ± (0.21)	Nuts	4.9 ± 0.32)		
Honey	2.1 ± (0.22)	Hot spicy	2.5 ± (0.01)		
Polen	2.1 ± (0.03)	Rancid	2.5 ± (0.02)		
Rancid	5.2 ± (0.04)	Salty	2.5 ± (0.25)		
Earthy	6.0 ± (0.03) *	Healthy	2.5 ± (0.01)		
		Plants	2.5 ± (0.01)		
		Red wine	1.2 ± (0.03)		

* Descriptors selected by consensus, with a citation frequency equal to or greater than 6% ± (S.D) indicated with an asterisk.

**Table 3 foods-13-03447-t003:** Loss of sensory characteristics applying the 0–3 scale (0: null, 1: slight, 2: moderate, 3: severe) for 8 species of wild edible mushrooms stored in a refrigerator at 4 °C for 8 days (method I). and scalded and frozen (method II). Samples were analyzed in triplicate. ^A–E^ Different letters indicate significant differences (*p* < 0.05) between different species for the same time. ^a–d^ Different letters indicate significant differences (*p* < 0.05) between different times for the species. Each method (I and II) was statistically analyzed separately.

	Method I				Method II		
	Time (Days)				Time (Months)		
Species	2	4	6	8	4	6	8
*G. vitellinum*	0.00 ± (0.00) ^B,c^	0.70 ± (0.15) ^E,b^	1.70 ± (0.15) ^B,a^	-------------	0.00 ± (0.00) ^B,c^	0.50 ± (0.17) ^A,c^	1.50 ± (0.17) ^B,a^
*P. dusenii*	0.00 ± (0.00) ^B,d^	0.50 ± (0.17) ^E,c^	1.50 ± (0.17) ^AB,b^	2.50 ± (0.17) ^A,a^	0.00 ± (0.00) ^B,c^	0.70 ± (0.15) ^A,c^	1.70 ± (0.15) ^B,a^
*C. hariotii*	0.00 ± (0.00) ^B,c^	1.00 ± (0.01) ^D,b^	1.90 ± (0.10) ^A,a^	--------------	0.00 ± (0.00) ^B,c^	0.70 ± (0.15) ^A,c^	1.70 ± (0.15) ^B,a^
*C. magellanicus*	1.00 ± (0.01) ^A,b^	2.60 ± (0.16) ^A,a^	---------------	--------------	0.40 ± (0.16) ^AB,c^	1.40 ± (0.16) ^B,b^	2.60 ± (0.16) ^A,a^
*F. antarctica*	1.00 ± (0.01) ^A,b^	2.20 ± (0.13) ^B,a^	---------------	--------------	0.50 ± (0.17) ^AB,c^	1.40 ± (0.16) ^B,b^	2.50 ± (0.17) ^A,a^
*F. endoxantha*	0.00 ± (0.00) ^B,b^	1.80 ± (0.13) ^C,a^	---------------	--------------	0.80 ± (0.25) ^B,b^	2.00 ± (0.21) ^A,a^	----------------
*R. patagonica*	0.00 ± (0.00) ^B,d^	0.50 ± (0.17) ^E,c^	1.50 ± (0.17) ^B,b^	2.50 ± (0.17) ^A,a^	2.20 ± (0.20) ^A^	----------------	----------------
*L. nuda*	0.00 ± (0.00) ^B,c^	0.50 ± (0.17) ^E,b^	1.50 ± (0.17) ^B,a^	--------------	0.00 ± (0.00) ^B,c^	0.70 ± (0.13) ^A,c^	1.70 ± (0.15) ^B,a^

**Table 4 foods-13-03447-t004:** List of VOCs detected in three mushroom species detected by SPME-GC-MS. RT = retention time, RIexp = Retention Index experimental, RIlit = Retention Index Literature database NIST. The odor descriptor was selected by Flavornet and The Good Scents company website. *m*/*z* corresponds to the three main fragment masses of the compound. VOC values are given in relative percentage ± SD. Samples were analyzed in triplicate.

N^o^	Name	CAS	RT	RIExp	RILit	Odor Descriptor	*m*/*z*			*R. patagonica*	*G. vitellinus*	*F. antartica*
1	Carbon dioxide	124-38-9	1.524	588	-		44	40	44	0.46 ± 0.38	1.59 ± 0.25	2.09 ± 1.18
2	Acetaldehyde	75-07-0	1.599	593	-	pungent, ethereal	44	43	42	0.3 ± 0.31	4.05 ± 1.48	0.61 ± 0.16
3	Ethanol	64-17-5	1.694	600	459	sweet	45	43	47	0.76 ± 1.08	1.28 ± 0.53	1.49 ± 2.11
4	2-propanone	67-64-1	1.749	604	-		43	58	42	0.20 ± 0.04	2.65 ± 0.83	1.05 ± 0.35
5	Acetic acid	64-19-7	2.094	630	628	sour	43	45	60	0.18 ± 0.25	nd	nd
6	butanal-3-methyl	590-86-3	2.569	666	659	malt	41	44	39	0.18 ± 0.12	0.28 ± 0.02	2.55 ± 0.04
7	butanal-2-methyl	96-17-3	2.679	674	671	cocoa, almond	57	86	41	0.04 ± 0.05	0.08 ± 0.03	0.93 ± 0.45
8	1-methoxy-2-propanol	107-98-2	2.749	679	673		45	43	47	1.32 ± 1.75	0.06 ± 0.09	1.73 ± 0.23
9	Pentanal	110-62-3	2.994	698	697	almond, malt, pungent	44	41	58	0.79 ± 0.01	0.28 ± 0.12	0.13 ± 0.06
10	2-methyl-pentenal	123-15-9	3.769	741	-		43	58	57	0.78 ± 0.22	0.02 ± 0.02	0.08 ± 0.04
11	3-methyl-3-pentanol	77-74-7	3.86	746	-		54	43	73	nd	0.08 ± 0.03	0.15 ± 0.06
12	2,3-butanediol	513-85-9	4.395	775	779	fruit, onion	45	43	57	0.01 ± 0.01	0.07 ± 0.05	3.13 ± 1.11
13	2-octene	111-67-1	4.67	790	810		43	41	55	0.16 ± 0.11	0.47 ± 0.4	nd
14	Hexanal	66-25-1	4.84	800	800	grass, tallow, fat	44	56	41	17.2 ± 0.33	1.43 ± 0.46	1.67 ± 0.11
15	2,4-dimethyl-heptane	2213-23-2	5.375	818	818		43	85	57	1.08 ± 1.52	nd	nd
16	1,3-octadiene	1002-33-1	5.47	822	826		54	67	41	0.18 ± 0.25	0.76 ± 0.21	0.09 ± 0.12
17	4-methyl-octane	2216-34-4	6.586	861	964		43	41	85	1.48 ± 2.06	2.40 ± 0.65	0.79 ± 1.11
18	3-methyl-butanoic acid	503-74-2	6.661	864	868	sweat, acid, rancid	60	41	43	0.14 ± 0.20	nd	0.74 ± 1.05
19	1-hexanol	111-27-3	6.711	865	870	resin, flower, green	56	43	42	0.44 ± 0.17	nd	0.17 ± 0.04
20	2-methyl-butanoic acid	116-53-0	7.006	876	868		74	41	57	0.06 ± 0.05	0.03 ± 0.02	0.05 ± 0.04
21	4-methyl-2-hexanone	105-42-0	7.346	888	-		43	58	41	0.14 ± 0.20	0.02 ± 0.03	0.17 ± 0.13
22	Heptenal	111-71-7	7.686	900	899	fat, citrus, rancid	43	41	70	1.05 ± 0.65	0.09 ± 0.02	0.25 ± 0.15
23	Butyrolactone	96-48-0	7.951	907	908		42	41	86	0.33 ± 0.46	0.35 ± 0.50	0.15 ± 0.22
24	Anisole	100-66-3	8.201	914	-		108	65	78	0.04 ± 0.06	nd	nd
25	Methyl hexanoate	106-70-7	8.501	923	924	fruit, fresh, sweet	74	43	87	1.08 ± 0.21	0.05 ± 0.02	nd
26	4-methyl-2-heptanone	6137-06-0	8.956	936	-		43	58	59	2.01 ± 1.89	0.58 ± 0.00	2.19 ± 0.66
27	Benzaldehyde	100-52-7	9.597	954	960	almond, burnt sugar	77	106	105	0.05 ± 0.07	0.09 ± 0.01	0.14 ± 0.02
28	4-methyl-nonane	17301-94-9	9.822	961	961		57	43	41	0.57 ± 0.80	0.90 ± 0.05	0.53 ± 0.14
29	2-methyl-nonane	871-83-0	9.917	964	966		57	71	43	2.14 ± 3.02	3.02 ± 0.19	1.91 ± 0.60
30	3-octanone	106-68-3	10.657	985	984	Mushroom, herb, butter, resin	55	43	71	0.57 ± 0.81	2.67 ± 2.62	0.13 ± 0.08
31	2-penthylfuran	3777-69-3	10.862	991	993	green bean, butter	81	82	53	0.15 ± 0.13	0.06 ± 0.02	0.10 ± 0.03
32	3-octanol	589-98-0	11.022	995	993	moss, nut, mushroom	59	55	83	1.14 ± 1.62	0.25 ± 0.11	2.74 ± 0.52
33	Decane	124-18-5	11.202	1001	-		43	57	41	1.37 ± 0.43	1.26 ± 0.49	1.58 ± 0.24
34	Octanal	124-13-0	11.247	1002	1001	fat, soap, lemon, green	43	57	44	1.36 ± 0.57	0.89 ± 0.96	0.08 ± 0.11
35	2-ethyl-2-hexenal	645-62-5	11.392	1006	1010		55	41	39	0.48 ± 0.01	0.06 ± 0.03	0.01 ± 0.02
36	4-methyl-decane	2847-72-5	11.543	1010	-		43	71	57	1.25 ± 0.34	1.18 ± 0.20	0.94 ± 0.25
37	2,5-dimethyl-nonane	17302-27-1	11.928	1020	-		57	43	41	3.67 ± 1.22	3.65 ± 0.77	2.19 ± 0.77
38	2,6-dimethyl-nonane	17302-28-2	12.068	1024	1022		43	71	57	14.00 ± 0.75	8.93 ± 0.90	4.47 ± 2.31
39	Limonene	138-86-3	12.203	1028	1026	lemon, orange	68	93	79	0.39 ± 0.09	0.82 ± 0.58	1.29 ± 1.01
40	2-ethyl-1-hexanol	104-76-7	12.243	1029	1025		57	41	43	2.64 ± 0.43	0.75 ± 0.89	14.72 ± 4.87
41	3-octen-2-one	1669-44-9	12.568	1038	1036	nut, crushed bug	55	43	111	0.43 ± 0.37	0.03 ± 0.03	0.50 ± 0.06
42	(E) 3-octen-2-one	18402-82-9	12.538	1037	1034		55	43	111	0.81 ± 0.13	0.12 ± 0.06	0.56 ± 0.00
43	Undecane	1120-21-4	13.323	1058	-	alkane	57	43	71	nd	16.89 ± 2.68	6.67 ± 3.73
44	Dodecane	112-40-3	13.553	1065	-	alkane	43	57	71	9.59 ± 0.78	6.20 ± 0.59	5.00 ± 1.69
45	(E) 2-octen-1-ol	18409-17-1	13.648	1067	1064	soap, plastic	57	41	55	nd	0.01 ± 0.01	0.27 ± 0.01
46	1-octanol	111-87-5	13.758	1070	1070		41	43	55	0.01 ± 0.02	0.04 ± 0.06	0.06 ± 0.04
47	2,3-dimethyldecane	17312-44-6	14.694	1096	-		43	71	57	1.03 ± 0.00	0.71 ± 0.03	0.58 ± 0.21
48	Nonanal	124-19-6	14.924	1102	1102	fat, citrus, green	57	41	55	nd	1.04 ± 1.47	nd
49	Octen-1-ol-acetate	77149-68-9	15.269	112	-		43	99	54	nd	0.03 ± 0.04	nd
50	2H-Pyran-2-one	2381-87-5	16.775	1153	-		82	39	54	nd	0.01 ± 0.01	0.06 ± 0.03
51	2-methyl-undecane	7045-71-8	17.175	1164	1164		43	57	71	2.69 ± 0.1	2.27 ± 0.82	1.74 ± 0.35
52	2-decanone	693-54-9	18.135	1191	1192		58	43	71	0.52 ± 0.06	0.15 ± 0.02	0.35 ± 0.15
53	Tridecane	629-50-5	18.465	1200	-	alkane	57	43	71	0.67 ± 0.01	0.83 ± 0.24	0.65 ± 0.09
54	Ethyl-1-hexanol acetate	103-09-3	18.68	1206	-		43	70	57	2.68 ± 0.48	1.68 ± 0.36	2.69 ± 0.40
55	2,6-dimethyl-undecane	17301-23-4	18.966	1215	1216		57	43	71	3.59 ± 0.12	3.85 ± 1.35	3.25 ± 0.48
56	Tetradecane	629-59-4	20.056	1246	-	alkane	57	43	71	2.25 ± 0.51	1.36 ± 0.51	2.45 ± 0.53
57	Pentadecane	629-62-9	21.271	1282	-	alkane	57	71	43	10.1 ± 0.56	13.9 ± 3.88	15.61 ± 0.9
58	2-undecanone	112-12-9	21.622	1292	1291	orange, fresh, green	43	58	71	2.36 ± 0.47	2.23 ± 0.17	3.16 ± 0.57
59	4,6-dimethyl-dodecane	61141-72-8	21.737	1295	-		57	71	43	3.07 ± 0.60	3.20 ± 0.67	4.61 ± 0.76
60	(+)-Cuparene	16982-00-6	28.489	1508	1504		132	131	145	nd	4.29 ± 0.15	0.09 ± 0

## Data Availability

The original contributions presented in the study are included in the article/Appendix A, further inquiries can be directed to the corresponding authors.

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
