# Peer review of "Sensory Characteristics and Volatile Organic Compound Profile of Wild Edible Mushrooms from Patagonia, Argentina"

_foods, 2024, doi:10.3390/foods13213447_

Round 1
Reviewer 1 Report
Comments and Suggestions for Authors
The manuscript describes the sensory and volatile organic compounds analysis of wild edible mushrooms from Patagonia. Overall the manuscript was well written and clearly describes the results. The results were also appropriate supported by data and placed in the context of current understanding using appropriate literature.
However, the manuscript has two figures labeled figure 2. The third figure needs to be appropriately labeled.
In addition, while I agree that the third figure is a valuable addition to the manuscript, I do not feel it was adequately described in the text. It is not clear to me what is included in the figure. I understand that it is a subset of the observed compounds but it is not completely clear what the how the sensory descriptor is connect to the observed compound concentration. For example hexanal has a odor descriptor of grass, tallow, fat. So is the concentration of hexanal included as fat or grass or both. This section needs to be more clearly described.
Author Response
Comment 1: The manuscript describes the sensory and volatile organic compounds analysis of wild edible mushrooms from Patagonia. Overall the manuscript was well written and clearly describes the results. The results were also appropriate supported by data and placed in the context of current understanding using appropriate literature.
Response 1: Thank you for your comments of the work. We appreciate your comments, and we believe they will help us to increase the quality of this publication. Please, find that changes in the manuscript were marked activating the “Track changes” option
Comment 2: However, the manuscript has two figures labeled figure 2. The third figure needs to be appropriately labeled.
Response 2: Sorry for the mistake. The figure numbers have been checked.
Comment 3: In addition, while I agree that the third figure is a valuable addition to the manuscript, I do not feel it was adequately described in the text. It is not clear to me what is included in the figure. I understand that it is a subset of the observed compounds but it is not completely clear what the how the sensory descriptor is connect to the observed compound concentration. For example hexanal has a odor descriptor of grass, tallow, fat. So is the concentration of hexanal included as fat or grass or both. This section needs to be more clearly described.
Response 3: Thank you for your comment. In the manuscript it is included this sentence ‘In order to enhance this approach correlating VOCs and attributes, expert tasters training with standards and chemical compounds quantification would be required.’ It is very difficult to establish a correlation between VOCs and sensory attributes because depends on 1) VOCs concentration (we have only relative percentage); 2) odor activity values (OAV) for each VOC; 3) threshold detection for each panelist (the lowest concentration of a substance in a medium). In the moment that the study was made, all these steps were not possible. However, we could tentatively correlate the VOCs with the attributes with the information we have.
Anyway, we have changed some sections in order to improve the explanation of this correlation. We hope that the current version solves their doubts and fulfills your expectations. Additionally, we will be happy to address any further requirements, suggestions, or comments.
Reviewer 2 Report
Comments and Suggestions for Authors
This paper describes the analysis of wild edible mushrooms from Pategonia, Argentina. The analyses were performed on dried then rehydrated mushrooms for sensory analysis; refrigerated and blanched then frozen mushrooms for sensory shelf life analysis; and VOC analysis using freeze dried mushrooms.
Table 1 was particularly helpful.
For the sensory shelf life experiment, the text refers to loss of flavor characteristics as well as gaining "a putrid odor." Were all the putrid odors the same? Bad odors could include amines, sulfur compounds, aldehydes, or carboxylic acids that all have off putting odors. More detail on the kind of bad odor would be helpful here.
For the VOC study, the method for freeze drying the mushrooms was not described.
For Table 4, additional descriptors for fragrance are available from SigmaAldrich, OSHA, etc. Including as many odors in your profile would enhance the discussion.
I agree with removing "alkane" odor from the profile. Compared to other odors, alkane odors are easily overshadowed by stronger smells.
The discussion includes conversation about bioactive compounds and antioxidant properties of these mushrooms. I agree that these are worthy of additional study, but this should be move to the final, or penultimate paragraph. Discussing these items earlier is confusing as they are not part of the present study.
Sugar content is mentioned in the discussion of F. antarctica. No quantitative data for sugar content is presented. I agree that sweet smelling VOCs can contribute to this odor profile, but so can sugar content. It would be helpful to present literature sugar content data for the mushrooms here.
Author Response
Comment 1: This paper describes the analysis of wild edible mushrooms from Pategonia, Argentina. The analyses were performed on dried then rehydrated mushrooms for sensory analysis; refrigerated and blanched then frozen mushrooms for sensory shelf life analysis; and VOC analysis using freeze dried mushrooms.
Table 1 was particularly helpful.
Response 1: Thank you for your comments of the work. We appreciate your comments, and we believe they will help us to increase the quality of this publication. Please, find that changes in the manuscript were marked activating the “Track changes” option.
Comment 2: For the sensory shelf life experiment, the text refers to loss of flavor characteristics as well as gaining "a putrid odor." Were all the putrid odors the same? Bad odors could include amines, sulfur compounds, aldehydes, or carboxylic acids that all have off putting odors. More detail on the kind of bad odor would be helpful here.
Response 2: Thank for your comment. In this case, 'putrid odor' refers to any unpleasant smell that may arise during mushroom senescence. We have expanded the information in the manuscript to clarify this aspect.
Comment 3: For the VOC study, the method for freeze drying the mushrooms was not described.
Response 3: Thank for your comment. The information has been included in materials and methods section.
Comment 4: For Table 4, additional descriptors for fragrance are available from SigmaAldrich, OSHA, etc. Including as many odors in your profile would enhance the discussion.
Response 4: Thank you for your comment. In this case, Flavornet and The good scents company website have been used to explore the sensory attributes. We agree with these two websites, and we have included additional odors in the Table 4.
Comment 5: I agree with removing "alkane" odor from the profile. Compared to other odors, alkane odors are easily overshadowed by stronger smells.
Response 5: Thank you very much for your comment.
Comment 6: The discussion includes conversation about bioactive compounds and antioxidant properties of these mushrooms. I agree that these are worthy of additional study, but this should be move to the final, or penultimate paragraph. Discussing these items earlier is confusing as they are not part of the present study.
Response 6: Thank you very much for your comment. We have modified the discussion section. Please check it.
Comment 7: Sugar content is mentioned in the discussion of F. antarctica. No quantitative data for sugar content is presented. I agree that sweet smelling VOCs can contribute to this odor profile, but so can sugar content. It would be helpful to present literature sugar content data for the mushrooms here.
Response 7: Thank you very much for your comment. We have included the sugar content in those species with sweet odor.
Reviewer 3 Report
Comments and Suggestions for Authors
After reading the manuscript "Sensory characteristics and volatile organic compound profile of Wild Edible Mushrooms from Patagonia, Argentina", I realized that the manuscript showed in some parts the scientific rigour wanted, but in other parts I have missed it.
The authors have presented critical evaluation only in some paragraphs.
The references are not exactly current, besides the objective, material and methods have to be improved.
Thats why I have written some suggestions below in an attempt to improve the paper.
L.33- mycogastronomy - This word is used 3 times in the text, one even as a keyword. Are you sure about its application in the paper? Check definition, pls.
L.35 - organoleptic properties - I no longer notice this term at the sensory analysis events I attend. Take a look at it.
L.53- "low fat, low sodium" and L.55 " low cholesterol, low sodium," - Again. please, reorganize the paragraph.
L.63- As you emphasize that they would be wild, it seems to me that something is missing about the guarantee of their safety being edible and not poisonous.
L.76- As this is a study in the field of sensory evaluation, I strongly suggest standardizing the terms in the paper, for example: odor x aroma ; organoleptic x sensory
L.84- QDA is a ®. Warning, please.
L. 101- Write the objective directly in a paragraph without letters (a) (b) (c)
L.124- Are the pictures really taken by the authors?
When authors state that all mushrooms can be eaten, don't you think it's safer to have an author to support this statement?
L.131- Which water ? and the temperature ?
Were the mushrooms hygienized before serving them to the assessors? how ?
Were they cooked? I ask because the PCA has flavor.
How were the samples offered to the evaluators?
on plates ? with random codes ?
Were the mushrooms in whole or pieces? Were they cut? Were caps and stems offered? I ask because you have soft and hard in the PCA and as I have worked with mushrooms I know that there is a difference between the stem and the cap.
L.138- Which author did you follow ?
L.140- It seems to me that a lot of information is missing for the QDA® test. Have you made a panellist selection? Were they mushroom consumers? Were they trained, but able to analyze wild mushrooms?
L.149- between or among ?
For an ADQ test a lot of relevant information was not included, I suggest reading papers and improving your article. Has the project been submitted to an evaluation by a university ethics committee? Did it follow the Helsinki declaration? Please, enter the approval protocol number. What was the profile of the assessors ? age of the assessors ? Are the assessors usually consumers of this product ? Were the analyses performed in sensory booths ? Did the assessors receive water to rinse the taste buds ? Authorss, the sensory part needs to be improved, a lot of important details are missing. It can't remain like this. L.227- Table 2- dot instead of comma L.277- You've explored PCA very little, so much to take advantage of. For example, what did you think of the fact that only 49% was explained? What about the earthy smell? and many other important things... Did you not associate the PCA findings with the cuisine, which in my opinion would be the cherry on the cake? L.278-283 - The writing of these lines is usual if you insert ellipses in the PCA L.307 - Table 3- dot instead of comma. Table 3 - For the statistical analysis, I miss out the upper and lower case letters for comparing the means: species x time x method. L.326- Figure 2 again ? L.350 - Table 4- dot instead of comma.L.454- I think it's more interesting when the conclusion is split from the Discussion.
L.459- Do not quote authors, it is the conclusion of your study.
Your conclusion must answer all your objectives.
Comments on the Quality of English LanguageMinor editing of English language required.
Author Response
Reviewer 3
Comment 1: After reading the manuscript "Sensory characteristics and volatile organic compound profile of Wild Edible Mushrooms from Patagonia, Argentina", I realized that the manuscript showed in some parts the scientific rigour wanted, but in other parts I have missed it. The authors have presented critical evaluation only in some paragraphs. The references are not exactly current, besides the objective, material and methods have to be improved. Thats why I have written some suggestions below in an attempt to improve the paper.
Response 1: Thank you for your comments of the work. We appreciate your comments, and we believe they will help us to increase the quality of this publication. Please, find that changes in the manuscript were marked activating the “Track changes” option.
Comment 2: L.33- mycogastronomy - This word is used 3 times in the text, one even as a keyword. Are you sure about its application in the paper? Check definition, pls.
Response 2: Thank you for your comment. We have changed the word.
Comment 3: L.35 - organoleptic properties - I no longer notice this term at the sensory analysis events I attend. Take a look at it.
Response 3: Thank you for your comment. We have changed the word.
Comment 4: L.53- "low fat, low sodium" and L.55 " low cholesterol, low sodium," - Again. please, reorganize the paragraph.
Response 4: Thank you for your comment. The first low sodium refers to the mushroom content, however the second low sodium to a specific diet. Anyway, the sentence has been reorganized.
Comment 5: L.63- As you emphasize that they would be wild, it seems to me that something is missing about the guarantee of their safety being edible and not poisonous.
Response 5: As indicated in line 61, the mushrooms of this study are included in the Argentine Food Code. Beyond this, we have included in this line an activity that is carried out by our institute every fall and spring "training in identification and safe consumption."
Comment 6: L.76- As this is a study in the field of sensory evaluation, I strongly suggest standardizing the terms in the paper, for example: odor x aroma ; organoleptic x sensory
Response 6: Thank you for your comment. We agree and we have standardized the terms.
Comment 7: L.84- QDA is a ®. Warning, please.
Response 7: Sorry for the mistake. We have included it.
Comment 8: L. 101- Write the objective directly in a paragraph without letters (a) (b) (c)
Response 8: Objectives have been separated in different paragraphs.
Comment 9: L.124- Are the pictures really taken by the authors?
Response 9: Yes, we have included this information in the Table 1 description.
Comment 10: When authors state that all mushrooms can be eaten, don't you think it's safer to have an author to support this statement?
Response 10: Yes, we strongly agree. For that we have included the reference in the title of the Table 1 “The consumption recommendations are from the Arts. 1249 and 1250 of the Argentinean Food Code.
Comment 11: L.131- Which water ? and the temperature ?
Response 11: New information has been included in the manuscript (line 139). “They were rehydrated in mineralized water at room temperature (20-24 ºC) for 2 hours. The soaking water was then carefully discarded and another quick wash was performed to remove any remaining grit before sensory evaluation.”
Comment 12: Were the mushrooms hygienized before serving them to the assessors? how ?
Response 12: Answered in the previous item
Comment 13: Were they cooked? I ask because the PCA has flavor.
Response 13: No, mushrooms were not cooked, only rehydrated as previously detailed, for the sensorial description. It is important to mention that the samples were only tasted and then spit out by the panelist after the description. They were not swallowed.
Comment 14: How were the samples offered to the evaluators? on plates ? with random codes ?
Response 14: Samples were putted individually in small dishes and offered in random order within each of two replicates using the 0–10 scale.
Comment 15: Were the mushrooms in whole or pieces? Were they cut? Were caps and stems offered? I ask because you have soft and hard in the PCA and as I have worked with mushrooms I know that there is a difference between the stem and the cap.
Response 15: Ramaria, Fistulina, Cyttaria and Lepista were offered in slices, as they were dryied. Ramaria, Fistulina and Cyttaria have no stem, while Lepista stems were removed prior to drying, as its fibrous texture is unpleasnt. Panus, Cortinarius and Gloesoma were offered complete, as they have small and thin fruitbodies. Gloeosoma has no stem, while Cortinarius and Panus have soft and edible stems.
More detailed information about this issue was included in lines 135-137. Details about fruitbodies characteristics can be found in “Toledo et al. 2016. Hongos comestibles silvestres de los bosques nativos de la región Andino Patagónica”. Accesible in: https://ri.conicet.gov.ar/handle/11336/115715. We decided not to include this reference as we were already required to eliminate references from the authors.
Comment 16: L.138- Which author did you follow ?
Response 16: We have included a new reference
Comment 17: L.140- It seems to me that a lot of information is missing for the QDA® test. Have you made a panellist selection? Were they mushroom consumers? Were they trained, but able to analyze wild mushrooms?
Response 17 Missing information was added, clarifying the general methodological reference for this section. Please check from line 133 to observe the changes made.
All panelists gave their informed consent to participate in the study, and authors have already submitted to the journal the previous informed consent form gave and accepted by them; we stated that responses will be anonymous.
Comment 18: L.149- between or among ?
Response 18: We have revised it.
Comment 19: For an ADQ test a lot of relevant information was not included, I suggest reading papers and improving your article. Has the project been submitted to an evaluation by a university ethics committee? Did it follow the Helsinki declaration? Please, enter the approval protocol number. What was the profile of the assessors ? age of the assessors ? Are the assessors usually consumers of this product ? Were the analyses performed in sensory booths ? Did the assessors receive water to rinse the taste buds ? Authorss, the sensory part needs to be improved, a lot of important details are missing. It can't remain like this.
Response 19: The appropriate protocols for protecting the rights and privacy of all participants were used. CIEFAP Research Center gave the permission to conduct sensory panel research. Testers were trained following ISO 13301:2022 (sensory analysis: general guidance for measuring odor, flavor, and taste detection thresholds by a three alternative forced-choice procedure) and 5495:2006 (sensory analysis: initiation and training of assessors in the detection and recognition of odors). The study was conducted according to the ethics guidelines of Nagoya Protocol (Argentinian National Law N˚ 27246). No other specific additional procedure is mandatory for this kind of study in our country in the date of this research, regarding further ethical approval, as the Executive director of the institute informed to the journal.
The assessors were researchers, students and worker recruited from CIEFAP Research Center and University of Patagonia. They were between 25 to 55 years old. They were interested and familiar with mushroom consumption, although not familiar with the species described in this study, as they are novel for the general public.
The sessions were performed in a meeting room with kitchen facilities, tables and chairs, reserved for this activity at CIEFAP research Center.
Panelist receive water, crackers and carrots to rinse the taste, as included in the text.
Comment 20: L.227- Table 2- dot instead of comma
Response 20: Dots have been included in the Table 2.
Comment 21: L.277- You've explored PCA very little, so much to take advantage of. For example, what did you think of the fact that only 49% was explained? What about the earthy smell? and many other important things... Did you not associate the PCA findings with the cuisine, which in my opinion would be the cherry on the cake?
Response 21: Thank you for your comment. We agree, and the PCA information has been amplify in results section.
Comment 22: L.278-283 - The writing of these lines is usual if you insert ellipses in the PCA
Response 22: Thank you for your comment. We have now changed the Figure 2.
Comment 23: L.307 - Table 3- dot instead of comma. Table 3 - For the statistical analysis, I miss out the upper and lower case letters for comparing the means: species x time x method.
Response 23: Dots have been included in the Table 3. We also add the missing letters.
Comment 24: L.326- Figure 2 again?
Response 24: Figure 2 has changed.
Comment 25: L.350 - Table 4- dot instead of comma.
Response 25: Sorry for the mistake. All the tables included commas but now only dots are included in Tables.
Comment 26: L.454- I think it's more interesting when the conclusion is split from the Discussion.
Response 26: Conclusion section has been created.
Comment 27: L.459- Do not quote authors, it is the conclusion of your study.
Response 27: As the conclusion section has been included, the references have been removed.
Comment 28: Your conclusion must answer all your objectives.
Response 28: Thank you for your comment. Conclusion section has been modified.
Round 2
Reviewer 3 Report
Comments and Suggestions for Authors After another evaluation of the manuscript, I realized some improvement in the quality of the paper. The authors have accepted almost all of my requests, and those they did not accept, they have justified with confidence.They added more authors to better substantiate the methodology and corrected tables and graphs.
However, unfortunately the top part since page 1 became disfigured and all the button parts were cut off and I couldn't re-evaluate. That's a pity.
I'm sorry if I didn't make myself clear, but regarding the objectives, my suggestion was to “Write the objective directly in a paragraph without letters (a) (b) (c)”. I've been a reviewer for Foods for some years and I don't remember seeing this format.
Comment 9: L.124- Are the pictures really taken by the authors?
Answer 9: "Yes, we have included this information in the Table 1 description. "
UnfortunateIy, couldn't evaluate it, it was cut off.
Response 13: " ....It is important to mention that the samples were only tasted and then spit out by the panelist after the description. They were not swallowed."I think this is very valuable information, and it's important to include it in the text.
Table 3 is very difficult to evaluate, it is disfigured in the document that was inserted into the system.
I ask you to check the statistical analysis of table 3, you have species x time x method. Was there an interaction? The statistical analysis I see for this type of analysis has 2 letters. For example Ab Aa - Please, read papers with the same proposal to better understand what I'm trying to explain.
You kindly provided me with many important answers about sensory analysis, but I couldn't find this important information in the version I received. However, it may have been cut off at the bottom of the paper. Please review it.
English is always useful to ask a native speaker for a final appreciation.
Comments on the Quality of English LanguageMinor editing of English language required.
Author Response
Comment 1: After another evaluation of the manuscript, I realized some improvement in the quality of the paper. The authors have accepted almost all of my requests, and those they did not accept, they have justified with confidence.
They added more authors to better substantiate the methodology and corrected tables and graphs.
However, unfortunately the top part since page 1 became disfigured and all the button parts were cut off and I couldn't re-evaluate. That's a pity.
Response 1: Thank you very much for your comments. We appreciate your comments, and we believe they will help us to increase the quality of this publication. Please, find that changes in the manuscript were marked activating the “Track changes” option.
Comment 2: I'm sorry if I didn't make myself clear, but regarding the objectives, my suggestion was to “Write the objective directly in a paragraph without letters (a) (b) (c)”. I've been a reviewer for Foods for some years and I don't remember seeing this format.
Response 2: Thank you for your comment. We have written the objectives in a single paragraph
Comment 3: Comment 9: L.124- Are the pictures really taken by the authors?; Answer 9: "Yes, we have included this information in the Table 1 description. "; UnfortunateIy, couldn't evaluate it, it was cut off.
Response 3: We apologize for the visualization issues you had. We do not understand why the manuscript was cut off. We uploaded the manuscript in word and pdf version. We checked the documents, and they were legible. Anyway, we have reviewed this new version and Tables (the pictures included) are legible now.
Comment 4: Response 13: " ....It is important to mention that the samples were only tasted and then spit out by the panelist after the description. They were not swallowed."I think this is very valuable information, and it's important to include it in the text.
Response 4: Thank you for your comment. We included that information in the text.
Comment 5: Table 3 is very difficult to evaluate, it is disfigured in the document that was inserted into the system.
I ask you to check the statistical analysis of table 3, you have species x time x method. Was there an interaction? The statistical analysis I see for this type of analysis has 2 letters. For example Ab Aa - Please, read papers with the same proposal to better understand what I'm trying to explain.
Response 5: As mentioned above, we have revised the document, and we hope you can reevaluate it. Regarding the statistical analysis, we have included a comparison between mushroom species for the same time and the different time for the same mushroom species. However, for a comparison between both methods (I and II) a direct conversion of time units would misrepresent the biological processes involved since they operate on different temporal scales (days and months, respectively). Each method reflects its temporal dynamics, and assuming linearity between these scales could distort the results. To preserve the integrity of the data and avoid introducing bias, we opted to analyze each method separately. Our focus is on understanding species responses within the temporal framework of each method, rather than forcing comparisons across different scales, ensuring the biological accuracy and reliability of the conclusions.
Comment 6: You kindly provided me with many important answers about sensory analysis, but I couldn't find this important information in the version I received. However, it may have been cut off at the bottom of the paper. Please review it.
Response 6: Please revise the new version.
Comment 7: English is always useful to ask a native speaker for a final appreciation.
Response 7: The English has been reviewed, and we hope the new version fully meets your expectations